# A Meta-MDP Approach to Exploration for Lifelong Reinforcement Learning

**Francisco M. Garcia and Philip S. Thomas**
College of Information and Computer Sciences
University of Massachusetts Amherst
Amherst, MA, USA
{fmgarcia,pthomas}@cs.umass.edu

## Abstract

In this paper we consider the problem of how a reinforcement learning agent that is tasked with solving a sequence of reinforcement learning problems (a sequence of Markov decision processes) can use knowledge acquired early in its lifetime to improve its ability to solve new problems. We argue that previous experience with similar problems can provide an agent with information about how it should *explore* when facing a new but related problem. We show that the search for an optimal exploration strategy can be formulated as a reinforcement learning problem itself and demonstrate that such strategy can leverage patterns found in the structure of related problems. We conclude with experiments that show the benefits of optimizing an exploration strategy using our proposed framework.

## 1 Introduction

One hallmark of human intelligence is our ability to leverage knowledge collected over our lifetimes when we face a new problem. When dealing with a new problem related to one we already know how to address, we leverage the experience obtained from solving the former problem. For example, upon buying a new car, we do not re-learn from scratch how to drive a car, instead we use the experience we had driving a previous car to quickly adapt to the new control and dynamics.

Standard *reinforcement learning* (RL) methods lack this ability. When faced with a new problem—a new *Markov decision process* (MDP)—they typically start learning from scratch, initially making uninformed decisions in order to *explore* and learn about the current problem they face. The problem of creating agents that can leverage previous experiences to solve new problems is called *lifelong learning* or *continual learning*, and is related to the problem of *transfer learning*.

Although the idea of how an agent can learn to learn has been explored for quite some time [14, 15], in this paper we focus on one aspect of lifelong learning: when faced with a sequence of MDPs sampled from a distribution over MDPs, how can a reinforcement learning agent learn an optimal policy for exploration? Specifically, we do not consider the question of *when* an agent should explore or *how much* an agent should explore, which is a well studied area of reinforcement learning research, [20, 10, 1, 5, 13]. Instead, we study the question of, given that an agent decides to explore, **which action should it take?** In this work we formally define the problem of searching for an *optimal exploration policy* and show that this problem can itself be modeled as an MDP. This means that the task of finding an optimal exploration strategy for a learning agent can be solved by another reinforcement learning agent that is solving a new *meta-MDP*, which operates at a different timescale from the RL agent solving a specific task—one episode of the meta-MDP corresponds to an entire lifetime of the RL agent. This difference of timescales distinguishes our approach from previous meta-MDP methods for optimizing components of reinforcement learning algorithms, [21, 9, 22, 8, 3].

We contend that ignoring the experience an agent might have with related MDPs is a missed opportunity for guiding exploration on novel but related problems. One such example is exploration by random action selection (as is common when using Q-learning, [23], Sarsa, [19], and DQN, [11]). To address this limitation, we propose separating the policies that define the agent's behavior into an *exploration* policy (which is trained across many related MDPs) and an *exploitation* policy (which is trained for each specific MDP).

In this paper we make the following contributions: **1)** we formally define the problem of searching for an optimal exploration policy, **2)** we prove that this problem can be modeled as a new MDP and describe one algorithm for solving this meta-MDP, and **3)** we present experimental results that show the benefits of our approach. Although the search for an optimal exploration policy is only one of the necessary components for lifelong learning (along with deciding *when* to explore, how to represent data, how to transfer models, etc.), it provides one key step towards agents that leverage prior knowledge to solve challenging problems. Code used for this paper can be found at https://github.com/fmaxgarcia/Meta-MDP

## 2   Related Work

There is a large body of work discussing the problem of *how* an agent should behave during exploration *when faced with a single MDP*. Simple strategies, such as $\epsilon$-greedy with random action-selection, *Boltzmann action-selection* or *softmax action-selection*, make sense when an agent has no prior knowledge of the problem that is currently trying to solve. It is a well-known fact that the performance of an agent exploring with unguided exploration techniques, such as random action-selection, reduces drastically as the size of the state-space increases [24]; for example, the performance of Boltzmann or softmax action-selection hinges on the accuracy of the action-value estimates. When these estimates are poor (e.g., early during the learning process), it can have a drastic negative effect on the overall learning ability of the agent. Given this limitation of unguided technique, when there is information available to guide an agent's exploration strategy, it should be exploited.

There exists more sophisticated methods for exploration; for example, it is possible to use state-visitation counts to encourage the agent to explore states that have not been frequently visited [20, 10]. Recent research has also shown that adding an exploration "bonus" to the reward function can be an effective way of improving exploration; VIME [6] takes a Bayesian approach by maintaining a model of the dynamics of the environment, obtaining a posterior of the model after taking an action, and using the KL divergence between these two models as a bonus. The intuition behind this approach is that encouraging actions that make large updates to the model allows the agent to better explore areas where the current model is inaccurate. [12] define a bonus in the reward function by adding an intrinsic reward. They propose using a neural network to predict state transitions based on the action taken and provide an intrinsic reward proportional to the prediction error. The agent is therefore encouraged to make state transitions that are not modeled accurately.

Another relevant work in exploration was presented in [3], where the authors propose building a library of policies from prior experience to explore the environment in new problems more efficiently. These techniques can be efficient when an agent is dealing with a single MDP; however, when facing a new problem they ignore potentially useful information the agent may have discovered from solving a previous task. That is, they fail to leverage prior experience. We aim to address this limitation by exploiting existing knowledge specifically for exploration. We do so by taking a meta-learning approach, where a meta-agent learns a policy that is used to guide an RL agent whenever it decides to explore, and contrast the performance of our method with Model Agnostic Meta-Learning (MAML), a state-of-the-art general meta-learning method which was shown to be capable of speeding up learning in RL tasks [4].

## 3   Background

A *Markov decision process* (MDP) is a tuple, $M = (\mathcal{S}, \mathcal{A}, P, R, d_0)$, where $\mathcal{S}$ is the set of possible states of the environment, $\mathcal{A}$ is the set of possible actions that the agent can take, $P(s, a, s')$ is the probability that the environment will transition to state $s' \in \mathcal{S}$ if the agent takes action $a \in \mathcal{A}$ in state $s \in \mathcal{S}$, $R(s, a, s')$ is a function denoting the reward received after taking action $a$ in state $s$ and transitioning to state $s'$, and $d_0$ is the initial state distribution. We use $t \in \{0, 1, 2, \dots, T\}$ to index the time-step, and write $S_t$, $A_t$, and $R_t$ to denote the state, action, and reward at time $t$. We also consider the *undiscounted* episodic setting, wherein rewards are not discounted based on the time

at which they occur. We assume that $T$, the maximum time step, is finite, and thus we restrict our discussion to *episodic* MDPs. We use $I$ to denote the total number of episodes the agent interacts with an environment. A *policy*, $\pi : \mathcal{S} \times \mathcal{A} \to [0, 1]$, provides a conditional distribution over actions given each possible state: $\pi(s, a) = \Pr(A_t = a | S_t = s)$. Furthermore, we assume that for all policies, $\pi$, (and all tasks, $c \in \mathcal{C}$, defined later) the expected returns are normalized to be in the interval $[0, 1]$.

One of the key challenges within RL, and the one this work focuses on, is related to the *exploration-exploitation dilemma*. To ensure that an agent is able to find a good policy, it should act with the sole purpose of gathering information about the environment (exploration). However, once enough information is gathered, it should behave according to what it believes to be the best policy (exploitation). In this work, we separate the behavior of an RL agent into two distinct policies: an *exploration* policy and an *exploitation* policy. We assume an $\epsilon$-greedy exploration schedule, i.e., with probability $\epsilon_i$ the agent explores and with probability $1 - \epsilon_i$ the agent exploits, where $(\epsilon_i)_{i=1}^I$ is a sequence of exploration rates where $\epsilon_i \in [0, 1]$ and $i$ refers to the episode number in the current task. We note that more sophisticated decisions on *when* to explore are certainly possible and could exploit our proposed method. Assuming this exploration strategy the agent forgoes the ability to learn when it should explore and we assume that the decision as to whether the agent explores or not is random. That being said, $\epsilon$-greedy is currently widely used (e.g.,SARSA [19], Q-learning [23], DQN [11]) and its popularity makes its study still relevant today.

Let $\mathcal{C}$ be the set of all tasks, $c = (\mathcal{S}, \mathcal{A}, P_c, R_c, d_0^c)$. That is, all $c \in \mathcal{C}$ are MDPs sharing the same state-set $\mathcal{S}$ and action set $\mathcal{A}$, which may have different transition functions $P_c$, reward functions $R_c$, and initial state distributions $d_0^c$. An agent is required to solve a set of tasks $c \in \mathcal{C}$, where we refer to the set $\mathcal{C}$ as the *problem class*. Given that each task is a separate MDP, the exploitation policy might not directly apply to a novel task. In fact, doing this could hinder the agent's ability to learn an appropriate policy. This type of scenarios arise, for example, in control problems where the policy learned for one specific agent will not work for another due to differences in the environment dynamics and physical properties. As a concrete example, Intelligent Control Flight Systems (ICFS) is an area of study that was born out of the necessity to address some of the limitations of PID controllers; where RL has gained significant traction in recent years [26, 27]. One particular scenario were our proposed problem would arise is in using RL to control autonomous vehicles [7], where a single control policy would likely not work for a number of distinct vehicles and each policy would need to be adapted to the specifics of each vehicle.

In our framework, the agent has a task-specific policy, $\pi$, that is updated by the agent's own learning algorithm. This policy defines the agent's behavior during exploitation, and so we refer to it as the *exploitation policy*. The behavior of the agent during exploration is determined by an *advisor*, which maintains a policy, $\mu$, tailored to the problem class (i.e., it is shared across all tasks in $\mathcal{C}$). We refer to this policy as an *exploration policy*. The agent is given $K = IT$ time-steps of interactions with each of the sampled tasks. Hereafter we use $i$ to denote the index of the current episode on the current task, $t$ to denote the time step within that episode, and $k$ to denote the number of time steps that have passed on the current task, i.e., $k = iT + t$, and we refer to $k$ as the *advisor time step*. At every time-step, $k$, the advisor suggests an action, $U_k$, to the agent, where $U_k$ is sampled according to $\mu$. If the agent decides to explore at this step, it takes action $U_k$, otherwise it takes action $A_k$ sampled according to the agent's policy, $\pi$. We refer to an optimal policy for the agent solving a specific task, $c \in \mathcal{C}$, as an *optimal exploitation policy*, $\pi_c^*$. More formally: $\pi_c^* \in \operatorname*{argmax}_\pi \mathbf{E}\left[G | \pi, c\right]$, where $G = \sum_{t=0}^T R_t$ is referred to as the return. Thus, the agent solving a specific task is optimizing the standard expected return objective. From now on we refer to the agent solving a specific task as the *agent* (even though the advisor can also be viewed as an agent). We consider a process where a task $c \in \mathcal{C}$ is sampled from some distribution, $d_\mathcal{C}$, over $\mathcal{C}$. While the RL agent learns how to solve a few of these tasks, the advisor also updates its policy to guide the agent during exploration. Whenever the agent decides to explore, it uses an action provided by the *advisor* according to its policy, $\mu$.

## 4 Problem Statement

We define the performance of the advisor's policy, $\mu$, for a specific task $c \in \mathcal{C}$ to be $\rho(\mu, c) = \mathbf{E}\left[\sum_{i=0}^I \sum_{t=0}^T R_t^i \Big| \mu, c\right]$, where $R_t^i$ is the reward at time step $t$ during the $i^{\text{th}}$ episode. Let $C$ be a random variable that denotes a task sampled from $d_\mathcal{C}$. The goal of the advisor is to find an *optimal*

*exploration policy*, $\mu^*$, which we define to be any policy that satisfies:

$$\mu^* \in \arg\max_{\mu} \quad \mathbf{E}\left[\rho(\mu, C)\right]. \tag{1}$$

In intuitive terms, this objective seeks to maximize the area under the learning curve of an agent. Assuming a stable policy $\pi$ whose performance improves with training (the performance of the policy does not collapse), maximizing this objective implies that the agent is able to learn more quickly. Because no single policy can solve every task, the meta-agent learns to help the agent obtain an optimal policy but it does not learn a policy to solve any task in particular.

Unfortunately, we cannot directly optimize this objective because we do not know the transition and reward functions of each MDP, and we can only sample tasks from $d_{\mathcal{C}}$. In the next section we show that the search for an exploration policy can be formulated as an RL problem where the advisor is itself an RL agent solving an MDP whose environment contains both the current task, $c$, and the agent solving the current task.

## 5   A General Solution Framework

Our framework can be viewed as a meta-MDP—an MDP within an MDP. From the point of view of the agent, the environment is the current task, $c$ (an MDP). However, from the point of view of the advisor, the environment contains both the task, $c$, and the agent. At every time-step, the advisor selects an action $U$ and the agent an action $A$. The selected actions go through a selection mechanism which executes action $A$ with probability $1 - \epsilon_i$ and action $U$ with probability $\epsilon_i$ at episode $i$.

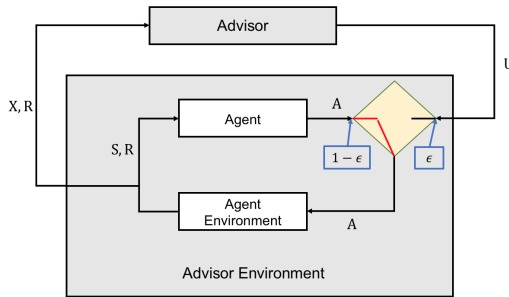

Figure 1: MDP view of interaction between the advisor and agent. At each time-step, the advisor selects an action $U$ and the agent an action $A$. With probability $\epsilon$ the agent executes action $U$ and with probability $1 - \epsilon$ it executes action $A$. After each action the agent and advisor receive a reward $R$, the agent and advisor environment transitions to states $S$ and $X$, respectively.

In our formulation, from the point of view of the advisor action $U$ is always executed and the selection mechanism is simply another source of uncertainty in the environment. Figure 1 depicts the proposed framework with action $A$ (exploitation) being selected. Even though one time step for the agent corresponds to one time step for the advisor, one episode for the advisor constitutes a lifetime of the agent. From this perspective, wherein the advisor is merely another reinforcement learning algorithm, we can take advantage of the existing body of work in RL to optimize the exploration policy, $\mu$.

We experimented training the advisor policy using two different RL algorithms: REINFORCE, [25], and Proximal Policy Optimization (PPO), [17]. Using Montercarlo methods, such as REINFORCE, results in a simpler implementation at the expense of a large computation time (each update of the advisor would require to train the agent for an entire lifetime). On the other hand, using temporal difference method, such as PPO, overcomes this computational bottleneck at the expense of larger variance in the performance of the advisor. Pseudocode for the implementations used in our framework using REINFORCE and PPO are shown in Appendix C.

### 5.1   Theoretical Results

Below, we formally define the meta-MDP faced by the advisor and show that an optimal policy for the meta-MDP optimizes the objective in (1). Recall that $R_c$, $P_c$, and $d_0^c$ denote the reward function, transition function, and initial state distribution of the MDP $c \in \mathcal{C}$. To formally describe the meta-MDP, we must capture the property that the agent can implement an arbitrary RL algorithm. To do so, we assume the agent maintains some memory, $M_k$, that is updated by some learning rule $l$ (an RL algorithm) at each time step, and write $\pi_{M_k}$ to denote the agent's policy given that its memory is $M_k$. In other words, $M_k$ provides all the information needed to determine $\pi_{M_k}$ and its update is of the form $M_{k+1} = l(M_k, S_k, A_k, R_k, S_{k+1})$ (this update rule can represent popular RL algorithms

like Q-Learning and actor-critics). We make no assumptions about which learning algorithm the agent uses (e.g., it can use Sarsa, Q-learning, REINFORCE, and even batch methods like Fitted Q-Iteration), and consider the learning rule to be unknown and a source of uncertainty.

**Proposition 1.** *Consider an advisor policy, $\mu$, and episodic tasks $c \in \mathcal{C}$ belonging to a problem class $\mathcal{C}$. The problem of learning $\mu$ can be formulated as an MDP, $M_{meta} = (\mathcal{X}, \mathcal{U}, T, Y, d'_0)$, where $\mathcal{X}$ is the state space, $\mathcal{U}$ the action space, $T$ the transition function, $Y$ the reward function, and $d'_0$ the initial state distribution.*

*Proof.* See Appendix A

$\square$

Given the formulated meta-MDP, $M_{meta}$, we are able to show that the optimal policy for this new MDP corresponds to an optimal exploration policy.

**Theorem 1.** *An optimal policy for $M_{meta}$ is an optimal exploration policy, $\mu^*$, as defined in* (1). *That is,*
$\mathbf{E}\left[\rho(\mu, C)\right] = \mathbf{E}\left[\sum_{k=0}^{K} Y_k \big| \mu, \right].$

*Proof.* See Appendix B.

$\square$

Since $M_{\text{meta}}$ is an MDP for which an optimal exploration policy is an optimal policy, it follows that the convergence properties of reinforcement learning algorithms apply to the search for an optimal exploration policy. For example, in some experiments the advisor uses the REINFORCE algorithm [25], the convergence properties of which have been well-studied [13].

Although conceptually simple, the framework presented thus far may require to solve a large number of tasks (episodes of the meta-MDP), each one potentially being an expensive procedure. To address this issue, we sampled a small number of tasks $c_1, \ldots, c_n$, where each $c_i \sim d_{\mathcal{C}}$ and train many episodes on each task in parallel. By taking this approach, every update to the advisor is influenced by several simultaneous tasks and results in an scalable approach to obtain a general exploration policy. In more difficult tasks, which might require the agent to train a long time, using TD techniques allows the advisor to improve its policy while the agent is still training.

# 6   Empirical Results

In this section we present experiments for discrete and continuous control tasks. Figures 8a and 8b depicts task variations for Animat for the case of discrete action set. Figures 11a and 11b show task variations for Ant problem for the case of continuous action set. Implementations used for the discrete case pole-balancing and all continuous control problems, where taken from OpenAI Gym, Roboschool benchmarks [2]. For the driving task experiments we used a simulator implemented in Unity by Tawn Kramer from the "Donkey Car" community [1]. We demonstrate that: **1)** in practice the meta-MDP, $M_{\text{meta}}$, can be solved using existing reinforcement learning methods, **2)** the exploration policy learned by the advisor improves performance on existing RL methods, on average, and **3)** the exploration policy learned by the advisor differs from the optimal exploitation policy for any task $c \in \mathcal{C}$, i.e., the exploration policy learned by the advisor is *not* necessarily a good exploitation policy. Intuitively, our method works well when there is a common pattern across tasks of what actions should *not* to be taken at a given state. For example, in a simple grid-world our method would not be able to learn a good exploration policy, but in the case of Animat (shown in figures 8a, and 8b) the meta-agent is able to learn that certain action patterns never lead to an optimal policy.

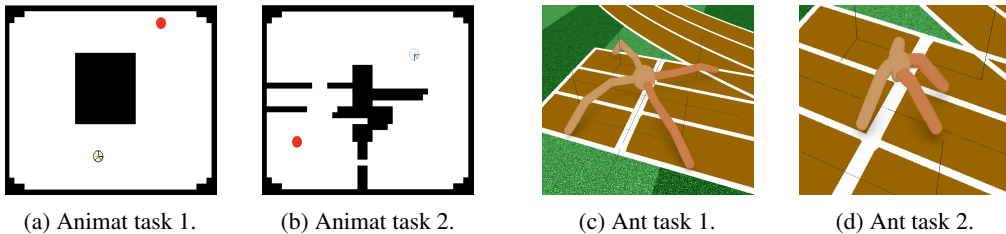

| (a) Animat task 1. | (b) Animat task 2. | (c) Ant task 1. | (d) Ant task 2. |

Figure 2: Example of task variations. The problem classes correspond to Animat (left) with discrete action space, and ant (right) with continuous action space.

To show that our algorithm behave as desired, we will first study the behavior of our method in two simple problem classes with discrete action-spaces: pole-balancing [19] and Animat [21], and a more realistic application of control tuning in self-driving vehicles. As a baseline meta-learning method, to which we contrast our framework, we chose Model Agnostic Meta Learning (MAML), [4], a general meta learning method for adapting previously trained neural networks to novel but related tasks. It is worth noting that, although the method was not specifically designed for RL, the authors describe some promising results in adapting behavior learned from previous tasks to novel ones.

## 6.1 Empirical Evaluation of Proposed Framework

We begin our evaluation by assessing the behavior of our algorithm in two different problems with discrete action spaces: Pole-balancing and Animat. We chose these problems because they present structural patterns that are intuitive to understand and can be exploited by the agent.

**Pole-balancing:** the agent is tasked with applying force to a cart to prevent a pole balancing on it from falling. The distinct tasks were constructed by modifying the length and mass of the pole mass, mass of the cart and force magnitude. States are represented by 4-D vectors describing the position and velocity of the cart, and angle and angular velocity of the pendulum, i.e., $s = [x, v, \theta, \dot{\theta}]$. The agent has 2 actions at its disposal; apply a force in the positive or negative $x$ direction. Figure 3a, contrasts the cumulative return of an agent using the advisor against random exploration during training over 6 tasks, shown in blue and red respectively. Both policies, $\pi$ and $\mu$, were trained using REINFORCE: $\pi$ for $I = 1,000$ episodes and $\mu$ for $500$ iterations. In the figure, the horizontal axis corresponds to episodes for the advisor. The horizontal red line denotes an estimate (with standard error bar) of the expected cumulative reward over an agent's lifetime if it samples actions uniformly when exploring. Notice that this is not a function of the training iteration, as the random exploration is not updated. The blue curve (with standard error bars from 15 trials) shows how the expected cumulative reward the agent obtains during its lifetime changes as the advisor improves its policy. After the advisor is trained, the agent is obtaining roughly $30\%$ more reward during its lifetime than it was when using a random exploration. To visualize this difference, Figure 3b shows the mean *learning curves* (episodes of an agent's lifetime on the horizontal axis and average return for each episode on the vertical axis) during the first and last 50 iterations.

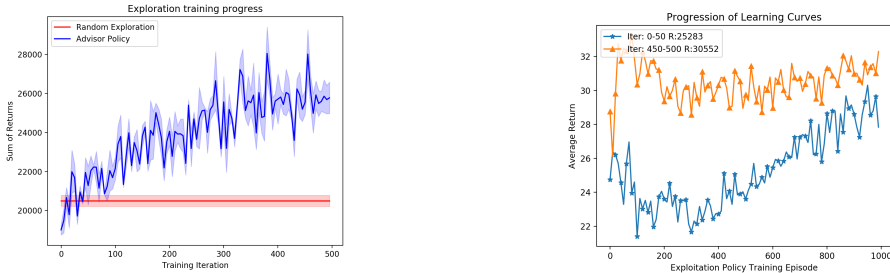

(a) Performance curves during training comparing advisor policy (blue) and random exploration policy (red).

(b) Average learning curves on training tasks over the first 50 advisor episodes (blue) and the last 50 advisor episodes (orange).

**Animat:** in these environments, the agent is a circular creature that lives in a continuous state space. It has 8 independent actuators, angled around it in increments of 45 degrees. Each actuator can be either on or off at each time step, so the action set is $\{0, 1\}^8$, for a total of 256 actions. When an actuator is on, it produces a small force in the direction that it is pointing. The resulting action moves the agent in the direction that results from the some of all those forces and is perturbed by 0-mean unit variance Gaussian noise. The agent is tasked with moving to a goal location; it receives a reward of $-1$ at each time-step and a reward of $+100$ at the goal state. The different variations of the tasks correspond to randomized start and goal positions in different environments. Figure 4a shows a clear performance improvement on average as the advisor improves its policy over 50 training iterations. The curve show the average curve obtained over the first and last 10 iteration of training the advisor, shown in blue and orange respectively. Each individual task was trained for $I = 800$ episodes.

An interesting pattern that is shared across all variations of this problem class is that there are actuator combinations that are not useful for reaching the goal. For example, activating actuators at opposite

angles would leave the agent in the same position it was before (ignoring the effect of the noise). The presence of these poor performing actions provide some common patterns that can be leveraged. To test our intuition that an exploration policy would exploit the presence of poor-performing actions, we recorded the frequency with which they were executed on unseen testing tasks when using the learned exploration policy after training and when using a random exploration strategy, over 5 different tasks. Figure 4b helps explain the improvement in performance. It depicts in the y-axis, the percentage of times these poor-performing actions were selected at a given episode, and in the x-axis the agent episode number in the current task. The agent using the advisor policy (blue) is encouraged to reduce the selection of known poor-performing actions, compared to a random action-selection exploration strategy (red).

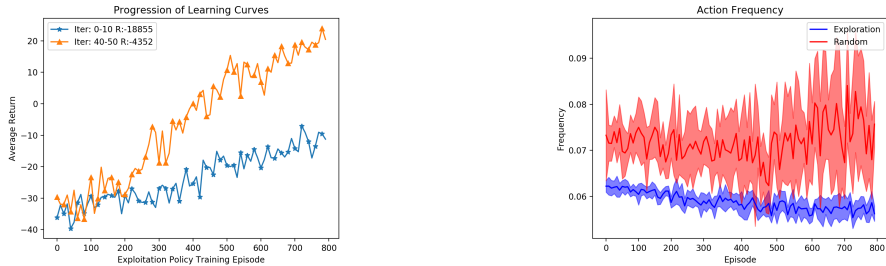

(a) Animat Results: Average learning curves on training tasks over the first 10 iterations (blue) and last 10 iterations (orange).

(b) Animat Results: Frequency of poor-performing actions in an agent's lifetime with learned (blue) and random (red) exploration.

**Vehicle Control:** a more pragmatic application of our framework is for quickly adapting control policy from one system to another. For this experiment, we tested the advisor on a control problem using a self-driving car simulator implemented in Unity. We assume that the agent has a constant acceleration (up to some maximum velocity) and the actions consist on $15$ possible steering angles between angles $\theta_{\min} < 0$ and $\theta_{\max} > 0$. The state is represented as a stack of the last $4$ $80 \times 80$ images sensed by a front-facing camera, and the tasks vary in the body mass, $m$, of the car and values of $\theta_{\min}$ and $\theta_{\max}$. We tested the ability of the advisor to improve fine-tuning controls to specific cars. We first learned a well-performing policy for one car and used the policy as a starting point to fine-tune policies for $8$ different cars.

The experiment, depicted in Figure 5, compares an agent who is able to use an advisor during exploration for fine-tuning (blue) vs. an agent who does not have access to an advisor (red). The figure shows the number of episodes of fine-tuning needed to reach a pre-defined performance threshold ($1,000$ time-steps without leaving the correct lane). The first and second groups in the figure show the average number of episodes needed to fine-tune in the first and second half of tasks, respectively. In the first half of tasks (left), the advisor seems to make fine-tuning more difficult since it has not been trained to deal with this specific problem. Using the advisor took an average of 42 episodes to fine-tune, while it took on average 12 episodes

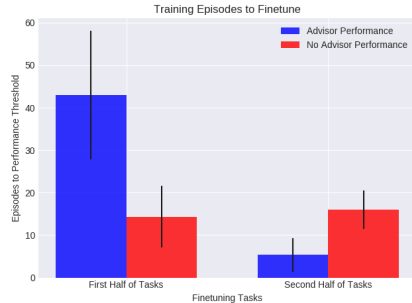

Figure 5: Number of episodes needed to achieve threshold performance (lower is better).

to fine-tune without it. The benefit, however, can be seen in the second half of training tasks. Once the advisor had been trained, it took on average 5 episodes to fine-tune while not using the advisor needed an average of 18 episodes to reach the required performance threshold. When the number of tasks is large enough and each episode is a time-consuming or costly process, our framework could result in important time and cost savings.

## 6.2 Is an Exploration Policy Simply a General Exploitation Policy?

One might be tempted to think that the learned policy for exploration might simply be a policy that works well in general. How do we know that the advisor is learning a policy for exploration and not simply a policy for exploitation? To answer this question, we generated three distinct unseen tasks

for pole-balancing and Animat problem classes and compared the performance of using only the learned exploration policy with the performance obtained by an exploitation policy trained to solve each specific task. Figure 6 shows two bar charts contrasting the performance of the exploration policy (blue) and the exploitation policy (green) on each task variation. In both charts, the first three groups of bars on the left correspond to the performance on each task and the last one to an average over all tasks. Figure 6a corresponds to the mean performance on pole-balancing and the error bars to the standard deviation; the y-axis denotes the return obtained. We can see that, as expected, the exploration policy by itself fails to achieve a comparable performance to a task-specific policy. The same occurs with the Animat problem class, shown in Figure 6b. In this case, the y-axis refers to the number of steps needed to reach the goal (smaller bars are better). In all cases, a task-specific policy performs significantly better than the learned exploration policy, indicating that the exploration policy is *not* a general exploitation policy.

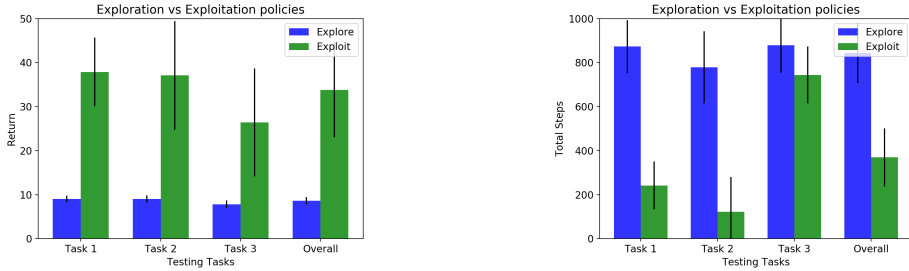

(a) Average returns obtained on test tasks when using the advisor's exploration policy (blue) and a task-specific exploitation (green)

(b) Number of steps needed to complete test tasks with advisor policy (blue) and exploitation (green).

Figure 6: Performance comparison of exploration and exploitation policies.

## 6.3 Performance Evaluation on Novel Tasks

We examine the performance of our framework on novel tasks when learning from scratch, and contrast our method to MAML trained using PPO. In the case of discrete action sets, we trained each task for 500 episodes and compare the performance of an agent trained with REINFORCE (R) and PPO, with and without an advisor. In the case of continuous tasks, we restrict our experiments to an agent using PPO after training for 500 episodes. In our experiments we set the initial value of $\epsilon$ to $0.8$, and decreased by a factor of $0.995$ every episode. The results shown in table 1 were obtained by training 5 times in 5 novel tasks and recording the average performance and standard deviations. The table displays the mean of those averages and the mean of the standard deviations recorded. The problem classes "pole-balance (d)" and "animat" correspond to discrete actions spaces, while "pole-balance (c)", "hopper", and "ant" are continuous.

| Problem Class | R | R+Advisor | PPO | PPO+Advisor | MAML |
|---|---|---|---|---|---|
| Pole-balance (d) | $20.32 \pm 3.15$ | $28.52 \pm 7.6$ | $27.87 \pm 6.17$ | $\mathbf{46.29 \pm 6.30}$ | $39.29 \pm 5.74$ |
| Animat | $-779.62 \pm 110.28$ | $\mathbf{-387.27 \pm 162.33}$ | $-751.40 \pm 68.73$ | $-631.97 \pm 155.5$ | $-669.93 \pm 92.32$ |
| Pole-balance (c) | — | — | $29.95 \pm 7.90$ | $\mathbf{438.13 \pm 35.54}$ | $267.76 \pm 163.05$ |
| Hopper | — | — | $13.82 \pm 10.53$ | $\mathbf{164.43 \pm 48.54}$ | $39.41 \pm 7.95$ |
| Ant | — | — | $-42.75 \pm 24.35$ | $83.76 \pm 20.41$ | $\mathbf{113.33 \pm 64.48}$ |

Table 1: Average performance (and standard deviations) over all unseen tasks trials on discrete and continuous control on the last 50 episodes.

## 7 Conclusion

In this work we developed a framework for leveraging experience to guide an agent's exploration in novel tasks, where the *advisor* learns the exploration policy used by the *agent* solving a task. We showed that a few sample tasks can be used to learn an exploration policy that the agent can use to improve the speed of learning on novel tasks. A takeaway from this work is that oftentimes an agent solving a new task may have had experience with similar problems, and that experience can be leveraged. One way to do that is to learn a better approach for exploring in the face of uncertainty. A natural future direction from this work use past experience to identify *when* exploration is needed and not just what action to take when exploring.

## Footnotes

[1]The Unity simulator for the self-driving task can be found at https://github.com/tawnkramer/sdsandbox

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
