[Supplementary Material]

# 8  Appendix A - Proof of Proposition 1

**Proposition 1.** *Consider an advisor policy, $\mu$, and episodic tasks $c \in \mathcal{C}$ belonging to a problem class $\mathcal{C}$. The problem of learning $\mu$ can be formulated as an MDP, $M_{meta} = (\mathcal{X}, \mathcal{U}, T, Y, d'_0)$, where $\mathcal{X}$ is the state space, $\mathcal{U}$ the action space, $T$ the transition function, $Y$ the reward function, and $d'_0$ the initial state distribution.*

*Proof.* To show that $M_{\text{meta}}$ is a valid MDP it is sufficient to characterize the MDP's state set, $X$, action set, $U$, transition function, $T$, reward function, $Y$, and initial state distribution $d'_0$. We assume that when facing a new task, the agent memory, $M$, is initialized to some fixed memory $M_0$ (defining a default initial policy and/or value function). The following definitions fully characterize the meta-MDP the advisor faces:

- $\mathcal{X} = \mathcal{S} \times \mathcal{I} \times \mathcal{C} \times \mathcal{M}$. That is, the state set $\mathcal{X}$ is a set defined such that each state, $x = (s, i, c, M)$ contains the current task, $c$, the current state, $s$, in the current task, the current episode number, $i$, and the current memory, $M$, of the agent.

- $\mathcal{U} = \mathcal{A}$. That is, the action set is the same as the action set of the problem class, $\mathcal{C}$.

- $T$ is the transition function, and is defined such that $T(x, u, x')$ is the probability of transitioning from state $x \in \mathcal{X}$ to state $x' \in \mathcal{X}$ upon taking action $u \in \mathcal{U}$. Assuming the underlying RL agent decides to explore with probability $\epsilon_i$ and to exploit with probability $1 - \epsilon_i$ at episode $i$, then $T$ is defined as follows:

$$
T(x, u, x') = \begin{cases} d_0^c(s')\mathbf{1}_{c'=c, i'=i+1, M'=l(M,s,a,r,s')} & \text{if } s \text{ is terminal and } i \neq I-1 \\ d_{\mathcal{C}}(c')d_0^{c'}(s')\mathbf{1}_{i'=0, M'=M_0} & \text{if } s \text{ is terminal and } i = I-1 \\ \left( \epsilon_i P_c(s, u, s') \right. \\ \left. + (1-\epsilon_i)\sum_{a \in A_c} \pi_M(s,a)P_c(s,a,s') \right) \\ \times \mathbf{1}_{c'=c, i'=i, M'=l(M,s,a,r,s')} & \text{otherwise.} \end{cases} \tag{2}
$$

- $Y$ is the reward function, and defines the reward obtained after taking action $u \in \mathcal{U}$ in state $x \in \mathcal{X}$ and transitioning to state $x' \in \mathcal{X}$. Notice that from the point of view of the meta-MDP, the reward function if a probability distribution and taking an action $u$ effectively samples from this distribution. Let $R$ be a random variable denoting the reward received by the meta agent, then $Y$ is given by:

$$
Y(x, u, x') = \begin{cases} \Pr(R = R_c(s, u, s')) = \epsilon_i + (1-\epsilon_i)\pi_M(u, s) \\ \Pr(R = R_c(s, a, s')) = (1-\epsilon_i)\,\pi_{Mk}(a, s), \forall a \in \mathcal{A}/\{u\} \end{cases} \tag{3}
$$

Also, notice that $\mathbf{E}\left[Y(x, u, x')\right] = \epsilon_i R_c(s, u, s') + (1-\epsilon_i)\sum_{a \in \mathcal{A}} \pi_M(a, s)R_c(s, a, s')$

- $d'_0$ is the initial state distribution and is defined by: $d'_0(x) = d_{\mathcal{C}}(c)d_0^c(s)\mathbf{1}_{i=0}$.

$\square$

# 9  Appendix B - Proof of Theorem 1

*Proof.* We can show that an optimal policy of $M_{\text{meta}}$ is an optimal exploration policy as defined in Eq. (1). To do so, it is sufficient to show that maximizing the return in the meta-MDP is equivalent to maximizing the expected performance. That is, $\mathbf{E}\left[\rho(\mu, C)\right] = \mathbf{E}\left[\sum_{k=0}^{K} Y_k \middle| \mu, \right]$.

$$
\mathbf{E}\left[\rho(\mu, C)\right] = \sum_{c \in \mathcal{C}} \Pr(C = c)\,\mathbf{E}\left[\sum_{i=0}^{I}\sum_{t=0}^{T} R_t^i \middle| \mu, C = c\right]
$$

$$= \sum_{c \in \mathcal{C}} \Pr(C = c) \sum_{i=0}^{I} \sum_{t=0}^{T} \mathbf{E}\left[R_t^i | \mu, C = c\right]$$

$$= \sum_{c \in \mathcal{C}} \Pr(C = c) \sum_{i=0}^{I} \sum_{t=0}^{T} \sum_{s \in \mathcal{S}} \Pr(S_{iT+t} = s | C = c, \mu) \, \mathbf{E}\left[R_t^i | \mu, C = c, S_{iT+t} = s\right]$$

$$= \sum_{c \in \mathcal{C}} \Pr(C = c) \sum_{i=0}^{I} \sum_{t=0}^{T} \sum_{s \in \mathcal{S}} \sum_{a \in \mathcal{A}} \Pr(S_{iT+t} = s | C = c, \mu)$$
$$\times \Pr(A_{iT+t} = a | S_{iT+t}, \mu) \, \mathbf{E}\left[R_t^i | \mu, C = c, S_{iT+t} = s, A_{iT+t} = a\right]$$

$$= \sum_{c \in \mathcal{C}} \Pr(C = c) \sum_{i=0}^{I} \sum_{t=0}^{T} \sum_{s \in \mathcal{S}} \sum_{a \in \mathcal{A}} \sum_{s' \in \mathcal{S}} \Pr(S_{iT+t} = s | C = c, \mu) \Pr(A_{iT+t} = a | S_{iT+t} = s, \mu)$$
$$\times \Pr(S_{iT+t+1} = s' | S_{iT+t} = s, A_{iT+t} = a, \mu)$$
$$\times \mathbf{E}\left[R_t^i | \mu, C = c, S_{iT+t} = s, A_{iT+t} = a, S_{iT+t+1} = s'\right]$$

$$= \sum_{c \in \mathcal{C}} \Pr(C = c) \sum_{i=0}^{I} \sum_{t=0}^{T} \sum_{s \in \mathcal{S}} \sum_{a \in \mathcal{A}} \sum_{s' \in \mathcal{S}} \Pr(S_{iT+t} = s | C = c, \mu) \Pr(A_{iT+t} = a | S_{iT+t} = s, \mu)$$
$$\times \left( \epsilon_i P_c(s, a, s') + (1 - \epsilon_i) \sum_{a' \in \mathcal{A}} \pi_{M_{iT+t}}(a', s) P_c(s, a', s') \right)$$
$$\times \left( \epsilon_i R_c(s, a, s') + (1 - \epsilon_i) \sum_{a' \in \mathcal{A}} \pi_{M_{iT+t}}(a', s) R_c(s, a', s') \right)$$

$$= \sum_{c \in \mathcal{C}} \Pr(C = c) \sum_{i=0}^{I} \sum_{t=0}^{T} \sum_{s \in \mathcal{S}} \sum_{a \in \mathcal{A}} \sum_{s' \in \mathcal{S}} \Pr(S_{iT+t} = s | C = c, \mu) \Pr(A_{iT+t} = a | S_{iT+t} = s, \mu)$$
$$\times T(x = (s, i, c, M_{iT+t}), a, x = (s', i, c, M_{iT+t}))$$
$$\times \mathbf{E}\left[Y(x = (s, i, c, M_{iT+t}), a, x = (s', i, c, M_{iT+t}))\right]$$

$$= \sum_{c \in \mathcal{C}} \Pr(C = c) \sum_{k=0}^{K} \sum_{s \in \mathcal{S}} \sum_{a \in \mathcal{A}} \sum_{s' \in \mathcal{S}} \Pr(S_k = s | C = c, \mu) \Pr(A_k = a | S_k = s, \mu)$$
$$\times T(x = (s, i, c, M_k), a, x' = (s', i, c, M_k))$$
$$\times \mathbf{E}\left[Y(x = (s, i, c, M_k), a, x' = (s', i, c, M_k)) | \mu, C = c, S_k = s, A_k = a, S_{k+1} = s'\right]$$

$$= \sum_{k=0}^{K} \sum_{x \in \mathcal{X}} \sum_{a \in \mathcal{A}} \sum_{x' \in \mathcal{X}} \Pr(X_k = x | \mu)$$
$$\times \Pr(U_k = a | X_k = x) T(x, u, x') \, \mathbf{E}\left[Y_k | X_k = s, U_k = a, X_{k+1} = x', \mu\right]$$

$$= \sum_{k=0}^{K} \sum_{x \in \mathcal{X}} \sum_{a \in \mathcal{A}} \sum_{x' \in \mathcal{X}} \Pr(X_k = x | \mu)$$
$$\times \Pr(U_k = a | X_k = x) \Pr(X_{k+1} = x' | U_k = a, X_k = x)$$
$$\times \mathbf{E}\left[Y_k | X_k = s, U_k = a, X_{k+1} = x', \mu\right]$$

$$= \sum_{k=0}^{K} \sum_{x \in \mathcal{X}} \sum_{a \in \mathcal{A}} \Pr(X_k = x | \mu) \, \Pr(U_k = a | X_k = x) \, \mathbf{E}\left[Y_k | X_k = x, U_k = a, \mu\right]$$

$$= \sum_{k=0}^{K} \sum_{x \in \mathcal{X}} \Pr(X_k = x | \mu) \, \mathbf{E}\left[Y_k | X_k = x, \mu\right]$$

$$= \sum_{k=0}^{K} \mathbf{E}\left[Y_k | \mu\right]$$

$$= \mathbf{E}\left[\sum_{k=0}^{K} Y_k | \mu\right]$$

□

# 10 Appendix C - Pseudocode

This section presents pseudocode for the REINFORCE and PPO implementation of the meta-MDP framework that were omitted from the paper for space considerations.

## 10.1 REINFORCE

Algorithm 1 presents pseudocode for an implementation of our method training the advisor using REINFORCE. The algorithm runs for $I_{meta}$ episodes for the advisor and $I$ episodes of the agent per advisor episode. At the end of each agent episode, the agent's policy $\pi$ parameterized by $\theta$ is updated via REINFORCE with step size $\alpha$, lines [10-12]. At the end of each advisor episode, every trajectory recorded by every agent episode is used to update the exploration policy $\mu$ parameterized by $\phi$ with REINFORCE using step size $\beta$, lines [13-15].

---
**Algorithm 1** Agent + Advisor - REINFORCE
---

    Initialize advisor policy $\mu$ randomly
    **for** $i_{meta} = 0, 1, \ldots, I_{meta}$ **do**
        Sample task $c$ from $d_c$
        **for** $i = 0, 1, \ldots, I$ **do**
            Initialize $\pi$ to $\pi_0$
            $s_t \sim d_0^c$
            **for** $t = 0, 1, \ldots, T$ **do**
                $a_t \sim \begin{cases} \mu \text{ with probability } \epsilon_i \\ \pi \text{ with probability } (1 - \epsilon_i) \end{cases}$
                take action $a_t$, observe $s_t$, $r_t$
            **for** $t = 0, 1, \ldots, T - 1$ **do**
                G $= \sum_{k=t+1}^{T} R_k$
                $\theta = \theta + \alpha\, G\, \nabla \log \pi(a_t, s_t)$
        **for** $t = 0, 1, \ldots, TI - 1$ **do**
            G $= \sum_{k=t+1}^{TI} R_k$
            $\phi = \phi + \beta\, G\, \nabla \log \mu(a_t, s_t)$

---

## 10.2 Proximal Policy Optimization (PPO)

Pseudocode for a PPO implementation of both agent and advisor is given in Algorithm 2. PPO maintains two parameterized policies for an agent, $\pi$ and $\pi_{old}$. The algorithm runs $\pi_{old}$ for $l$ time-steps and computes the generalized advantage estimates (GAE), [16], $\hat{A}_{s_1}, \ldots, \hat{A}_{s_l}$, where $\hat{A}_{s_t} = \delta_{s_t} + (\gamma\lambda)\delta_{s_{t+1}} + \cdots + (\gamma\lambda)^{l-t+1}\delta_{s_{l-1}}$ and $\delta_{s_t} = r_t + \gamma V(s_{t+1}) - V(s_t)$.

The objective function seeks to maximize the following objective for time-step $t$:

$$J = \mathbf{E}_t \left[ \min(r_t\hat{A}_t, \text{clip}(r_t, 1-\alpha, 1+\alpha)\hat{A}_t) \right]$$
$$- (R_t + \gamma\, \hat{V}(s_{t+1}) - \hat{V}(s_t))^2 \tag{4}$$

where $r_t = \frac{\pi(a_t|s_t)}{\pi_{old}(a_t|s_t)}$, and $\hat{V}(s)$ is an estimate of the value for state $s$. The updates are done in mini-batches that are stored in a buffer of collected samples.

To train the agent and advisor with PPO we defined two separate sets of policies: $\mu$ and $\mu_{old}$ for the advisor, and $\pi$ and $\pi_{old}$ for the agent. The agent collects samples of trajectories of length $l$ to update its policy, while the advisor collects trajectories of length $n$, where $l < n$. So, $J$ (the objective of the agent) is computed with $l$ samples while $J_{meta}$ (the objective of the advisor) is computed with $n$ samples. In our experiments, setting $n \geq 2l$ seemed to give the best results.

Notice that the presence of a buffer to store samples, means that the advisor will be storing samples obtained from many different tasks, which prevents it from over-fitting to one particular problem.

---

**Algorithm 2** Agent + Advisor - PPO

---

1: Initialize advisor policy $\mu$, $\mu_{old}$ randomly
2: **for** $i_{meta} = 0, 1, \ldots, I_{meta}$ **do**
3:     Sample task $c$ from $d_c$
4:     **for** $i = 0, 1, \ldots, I$ **do**
5:         Initialize $\pi$ and $\pi_{old}$
6:         $s_t \sim d_0^c$
7:         $x_t = (s_t, i, c)$
8:         **for** $t = 0, 1, \ldots, T$ **do**
9:             $a_t \sim \begin{cases} \mu_{old} \text{ with probability } \epsilon_i \\ \pi_{old} \text{ with probability } (1 - \epsilon_i) \end{cases}$
10:             take action $a_t$, observe $s_t, r_t$
11:             **if** $t \% l = 0$ **then**
12:                 compute $\hat{A}_{s_1}, \ldots, \hat{A}_{s_l}$
13:                 optimize $J$ w.r.t $\pi$
14:                 $\pi_{old} = \pi$
15:             **if** $t \% n = 0$ **then**
16:                 compute $\hat{A}_{x_1}, \ldots, \hat{A}_{x_n}$
17:                 optimize $J_{meta}$ w.r.t $\mu$
18:                 $\mu_{old} = \mu$

---

# 11 Appendix D - Task Variations

This section shows variations of each problem used for experiments.

**Pole-balancing:** Task variations in this problem class were obtained by changing the mass of the cart, the mass and the length of the pole.

(a) Pole-balancing task 1.

(b) Pole-balancing task 2.

Figure 7: Experiments 1 of task variations with discrete action space.

**Animat:** Task variations in this problem class were obtained by randomly sampling new environments, and changing the start and goal location of the Animat.

(a) Animat task 1.

(b) Animat task 2.

Figure 8: Experiments 2 of task variations with discrete action space.

**Driving Task:** Task variations in this problem class were obtained by changing the mass of the car and turning radius. A decrease in body mass and increase in turning radius causes the car to drift more and become more unstable when taking sharp turns.

(a) Driving task 1.

(b) Driving task 2.

Figure 9: Experiments 3 of task variations with discrete action space.

**Hopper:** Task variations for Hopper consisted in changing the length of the limbs, causing policies learned for other hopper tasks to behave erratically.

(a) Hopper task 1.

(b) Hopper task 2.

Figure 10: Experiments 1 of task variations with continuous action space.

**Ant:** Tasks variation for Ant consisted in changing the length of the limbs in the ant and the size of the body.

(a) Ant task 1.

(b) Ant task 2.

Figure 11: Experiments 2 of task variations with continuous action space.