[Reviews · NeurIPS 2019]

Reviewer 1



The authors propose a learning algorithm for adviser who solves Meta-MDP. During the learning of the agent who solves the specific task, the advisor advises the agent to take some action u and it is chosen with probability epsilon. The authors show the advantage of their method compared with MAML in the tasks Pole-balancing, Animat, and Vehicle Control. ------ Comments after Authors' feedback -------------- Thanks for your feedback. I still cannot get rid of my concern. So I will keep my score. > The exploration policy (the epsilon part) is the one learned by the advisor, which as shown in Figure 6, is not simply a general policy to solve many tasks. If my understanding is correct, what the authors compare with the proposed method is the task-specific exploitation policy. I believe it is much more reasonable to compare the adviser with the policy that is trained on all the past environments, which will presumably be robust to the change of the environment. I am wondering if such a simple domain randomized policy works as an adviser., and am also wondering if the proposed stochastic selection of the action is a really good method to convey the knowledge of the adviser. So I'd like to request the authors to compare the proposed method with the one that use a domain-randomized policy as an initial guess of new task to validate the proposed method actually works better than this simple baseline.

Reviewer 2



I am on the fence for this paper. I like that the approach is simple. I like that it was tested in different domains. I like that several different views of performance are included. Nevertheless I have many questions, that need resolution before I can increase my score. As it stands I am leaning to reject. The basic idea is to learn a separate exploration policy with an independent RL agent. The system does not learn when to explore, but how to explore assuming the main controller requests an exploratory action. There is surprising little work on learning separate exploration policies beyond the older work such as Simsek and Barto. I expected this paper to discuss Simseks work a bit, even if it was not deigned for function approximation. The motivation of learning an exploration policy from a lifetime is a good one. However, the paper goes on to assume tasks are stochastically drawn from a distribution and training only multiple tasks occurs in parallel. This is both unrealistic, and it also seems like it would limit the main idea. That training on an increasingly difficult sequence of tasks would result in better and better exploration. That one view of this approach. Another view is that by learning from a set of tasks, the agent would learn a policy that would be useful to explore in new domains. Unfortunately the paper does not directly test this. As far as I understand no experiment trains the architecture on a set of training tasks, then fixes the exploration policy, and then evaluates the system on a totally new task. In addition it would be nice to see some more analysis of the learned exploration behavior. Section 6.2 investigates this a bit, checking if the learned policy is simply an exploitative policy. What about investing how the advisor policy changes with time? I would like to see a bit more leg work to rule out other effects and better discussion of the results. Why not include a strong exploration method as a baseline, like Osband's Randomized Value Functions or [12]. That would give us a baseline to understand the results. Why not compare against a PPO agent that is not reset after each task---show that transfer fails! For instance, the issue described in section 6.1---that in animat some actuator combinations are not useful---could be discovered by a transfer agent. The results presentation could be improved. None of the figures have figure numbers. Some plots have error bars others don't. The plots with the most noisy lines and the biggest questions about significance have no error bars. In the animate results the range of the y-axis is very small. There are error bars but they are not the same size so it's difficult to conclude that the new method is clearly better. In table 1 some results are not statistically significant, but still bolded. Why is there no training progress plot for Animat like there was for pole balancing? In addition, important details are missing. What was the function approximator used in pole balancing? Why only 15 runs in pole-balancing? What are the state variables in Animat? Why is there an additional +100 at the goal in Animat---if this is a cost to goal problem this is not needed? How were any of the hyper-parameters tuned in any of the experiments. How do we know the comparisons agains PPO are fair without a clear description of how these critical performance parameters were set? I look forward to the author response! ================================ Update based on author response: Thank you for clarifying about the hold-out test experiment. There are still missing baselines, and open questions that prevent limit the work and were not directly addressed in the author response (but I understand space is limited). The reviewers had a nice discussion of what the results mean, and how they should be communicated. The other reviewer will summarize, but in short please be more clear describing how the experiments described in table 1. Please describe how significance was calculated and spend more time describing the results. This is a general comment for the presentation in the paper. Describe error bars when included. Explain why the results matter (sometimes the axis range on the graphs are tiny and large). Explain how baselines and your approach were tuned. Identify sources of bias. Help the reader understand the details of your results and why they are important---after all, the contribution of the paper is evaluated entirely empirically here! Nice to have: more analysis of the exploration policy and how it evolves over time. Something more contextual than action frequency. For example, could you design an interesting gridworld and visualize the exploration policies at the beginning and end of learning? Can you think of gridworlds that should cause the exploration policies to evolve differently during training? Finally, the way you generate different tasks is reminiscent of the work of Whiteson et al: "Protecting against evaluation overfitting in empirical reinforcement learning"

Reviewer 3



This paper is clearly written. It presents a new problem formulation and proceeds to examine it carefully. It is interesting to see that even when the agent follows an epsilon-greedy policy one can still do better than randomly selecting actions. It is also interesting to see that the learned exploration policy does indeed change during learning and it does improve agents' performance. To the best of my knowledge this problem formulation is new and it would actually be interesting to see this line of work being further explored. One thing I missed when reading the paper was reading more intuitions about the proposed approach. For example, why learning a policy that maximizes the AUC of an RL agent is not very correlated to the agent's exploitative performance? What were the behaviours it learned in CartPole and the self-driving car task? I really liked the discussion about how the exploration policy in Animats learns to avoid actions that cancel each other. There's no such phenomenon in the other tasks, as far as I know. How is this policy effective? The discussion is sometimes confusing. I was a little bit lost in Section 6.1, for example, when Figure 5 is explained. Finally, MAML is not the state-of-the-art in meta-learning. For example, REPTILE is an algorithm that directly builds on top of MAML and that leads to better performance (https://arxiv.org/abs/1803.02999). I'd expect a stronger baseline in some of the results. The paper would benefit from being proofread for English. 'where' and 'were' are often mixed up. There are some typos as well ("The curve show", "0.995 ever episode"). -- Comments after the authors' response -- This paper has a simple and interesting idea to tackle the problem of exploration in a sequence of similar tasks. The empirical analysis is interesting and broad. Thus, I'm keeping my recommendation for accepting this paper. I suggest the authors to clarify three things in this paper though: 1. What is the intuition behind the fact that maximizing the AUC across tasks won't lead to an exploitative policy? Is simply the fact that different tasks don't accept the same exploitative policy and the learned exploratory policy is kind of good but general? 2. Further explaining the properties of the learned exploratory policy would be interesting. 3. After the rebuttal and discussion something started to worry me: the bold numbers in Table 1. What does the bold mean? Is it a statistical confidence? What is the p value? *Importantly*, and I can't stress this enough, what was the N used? Was it N=5 (as it should be), or N=50? This changes the interpretation of the significance of Pole-balance (d) for the PPO experiments. Since N=5, was a non-parametric test used? Was it corrected for multiple comparisons? I didn't reject this paper based on Table 1 because across tasks the mean performance obtained when using the Advisor is consistently good, showing a general trend, but the bold numbers might be very misleading and I'd ask the authors to be careful with that.

[Author Response · NeurIPS 2019]

Thank you to all reviewers for taking the time to give us feedback on our submission. We hope the following addresses your concerns.

**Reviewer 1:**

- We would like to clarify, the *greedy* policy (exploitation) is not the one learned by the advisor. That is the task specific policy. The exploration policy (the epsilon part) is the one learned by the advisor, which as shown in Figure 6, is not simply a general policy to solve many tasks.

- We will expand our explanation on when we can expect this method to be useful. Intuitively, it helps when in a given state certain actions are not appropriate regardless of task variations. For example, if the pole in pole balancing is about to fall to the right, moving the cart to the left will never help the situation.

- Thank you for the recommendation on other exploration baselines. We will take them into account.

**Reviewer 2:**

- We also found it surprising how little work there is in this specific area. After going over your suggestion, we believe a discussion on Simsek and Barto's work is appropriate.

- We apologize if there was some confusion and we will clarify in the text. 1) Yes, we do assume that tasks come from a distribution for our theoretical formulation; however, in practice we use a finite set of tasks to solve, a small subset of which is used to learn the exploration policy. 2) No, we do not need to train in parallel. The results hold when training on different MDPs sequentially, but when possible, updating the advisor from multiple tasks in parallel makes learning an exploration policy quicker. Good point! Training on increasingly difficult problems, like in curriculum learning, would be a clear scenario where learning an exploration policy, as we propose, could lead to clear benefits.

- We did perform the experiment you are suggesting to evaluate the learned exploration policy. Figures 5 and table 1, evaluate the performance of the learned exploration policy on a set of novel tasks (different from the ones used for training), showing that it leads to clear performance improvement.

- Your comment on the Animat problem is a fair point. A simple transfer of policies would definitely reduce the number of 'non useful' actions taken. However, it would be highly biased to the task where the policy was learned, which could lead to really poor performance in a new task. We evaluated this point based on your comment. The figure on the right shows the (average) effect of a simple transfer of policy over five task variations. The right plot shows that the frequency with which the agent takes 'non useful' actions is decreased significantly with simple policy transfer; however, left plot shows that a simple transfer of policy can actually make learning more difficult.

Effect of Transfer on Animat

- We will gladly add your suggestions to improve presentation. The reason why no training progress is shown for Animat is that the plot would not show anything that was not shown for pole-balancing, and we thought that using the space to analyze the action selection process in Animat would be more relevant.

**Reviewer 3:**

- We will make sure to clarify on the intuition behind our objective. Assuming an agent improves the expected return of the policy with each episode, maximizing the cumulative return is equivalent to maximizing the AUC. Intuitively, when plotted, the quicker an agent reaches an optimal policy, the larger the AUC will be. Because the only variable that is optimized is the exploration policy, it learns to exploit structures present on the tasks to achieve this.

- The behaviors learned for pole balancing and self-driving task were a bit more intuitive. In pole balancing, the exploration policy learns that if the pole is about to fall to the right, the cart should compensate by moving to the right, and vice versa. Similarly, in the self-driving task, if the car is driving to much to the right the appropriate action should be moving to the left by some amount and vice versa. In general, certain actions can safely be omitted from exploration in certain states. We will make a more detailed discussion regarding this result.

- Figure 5 is showing that, in the first half of the set of tasks (while the advisor has not trained sufficiently), the exploration policy is inefficient. The second half of set of tasks demonstrate that the exploration policy is improving over the initial exploration policy and leads to improving learning over random exploration. We will clarify the points discussed in section 6.1

- Thank you for pointing out REPTILE as an alternative meta learning method.

[Meta-Review · NeurIPS 2019]

Even after the discussion and the author response there was still some disagreement between the reviewers. The paper proposes a simple yet novel and very interesting idea. There still are a few concerns about clarity, but those can be fixed in the final version (see updated reviews). Overall this is a solid paper, that (as always) would benefit from more thorough empirical evaluation. One reviewer proposed to add an additional baseline of a domain-randomized robust policy that is trained on various tasks. In summary an interesting idea that might not be a 100% fleshed out but is ready to be "put out there" nevertheless. It would be good to mention the AAMAS extended abstract in the final version and to discuss the relation to it.